# Blood Biomarkers for Triaging Patients for Suspected Stroke: Every Minute Counts

**DOI:** 10.3390/jcm11144243

**Published:** 2022-07-21

**Authors:** Radhika Kiritsinh Jadav, Reza Mortazavi, Kwang Choon Yee

**Affiliations:** 1Faculty of Health, University of Canberra, Canberra, ACT 2617, Australia; radhikashah2529@gmail.com (R.K.J.); kwang_choon.yee@canberra.edu.au (K.C.Y.); 2Prehab Activity Cancer Exercise Survivorship Research Group, Faculty of Health, University of Canberra, Canberra, ACT 2617, Australia

**Keywords:** stroke, CNS, ischaemic, haemorrhagic, biomarker, panel, young adults, children, triage, specificity, sensitivity, prediction values

## Abstract

Early stroke diagnosis remains a big challenge in healthcare partly due to the lack of reliable diagnostic blood biomarkers, which in turn leads to increased rates of mortality and disability. Current screening methods are optimised to identify patients with a high risk of cardio-vascular disease, especially among the elderly. However, in young adults and children, these methods suffer low sensitivity and specificity and contribute to further delays in their triage and diagnosis. Accordingly, there is an urgent need to develop reliable blood biomarkers for triaging patients suspected of stroke in all age groups, especially children and young adults. This review explores some of the existing blood biomarkers, as single biomarkers or biomarker panels, and examines their sensitivity and specificity for predicting stroke. A review was performed on PubMed and Web of Science for journal articles published in English during the period 2001 to 2021, which contained information regarding biomarkers of stroke. In this review article, we provide comparative information on the availability, clinical usefulness, and time-window periods of seven single blood biomarkers and five biomarker panels that have been used for predicting stroke in emergency situations. The outcomes of this review can be used in future research for developing more effective stroke biomarkers.

## 1. Introduction

Stroke is the leading cause of disability and the second most common cause of death worldwide [1]. Early detection of stroke is essential for implementing timely diagnostic tests and radio-imaging, as well as subsequent intervention therapies such as thrombolysis (using tissue plasminogen activator), thrombectomy, or anti-platelet/anti-coagulant treatments [2,3,4,5,6]. However, early detection of stroke is still remaining elusive, and it has been reported that even in many advanced hospitals, only about one-third of the patients with ischaemic stroke (IS) are diagnosed early enough for a timely intervention [2].

Early screening tools, such as the Cincinnati Prehospital Stroke Scale (CPSS) or the Recognition of Stroke in the Emergency Room (ROSIER) scale, have demonstrated their values in high-risk patients, with a sensitivity between 80% and 85% [2,7]. However, these tools are less accurate in children and young adults, who account for 10−15% of all stroke cases [2,8]. Given there are approximately 12 million new cases of stroke diagnosed globally each year, it is estimated that there are around 1–2 million cases per year that are not detected appropriately using the current screening tools [9]. In addition, studies have found that current screening tools have poor performance in distinguishing stroke from stroke mimics such as migraine, epilepsy, central nervous system (CNS) infections, Bell’s palsy, and conversion disorders, with a negative predictive value of approximately 20% [2,10].

The majority of the current screening tools for stroke are based on considering the patients’ clinical signs and symptoms and demographic risk factors. The downside is that those patients who do not present with typical symptoms and those who are perceived as low risk (e.g., children and young adults) may not be consistently identified [8]. Therefore, we need alternative methods for detecting potential stroke cases which do not depend on the above-mentioned categorisations. Such screening tools would be a welcome addition to the diagnostic toolkit of clinicians at emergency departments, neurology departments, and regional hospitals, as well as paramedics.

The use of blood biomarkers plays an important role in the screening and diagnosis of some critical illnesses such as ischaemic heart disease. The inclusion of troponin into the screening/diagnostic protocols of ischaemic heart disease in the early 2000s significantly improved the clinical approach to this condition, and subsequently has contributed to remarkable improvements in patient outcomes [11]. Unfortunately, this is not the case with the screening and diagnosis of stroke.

The brain is a complex tissue comprising different unique cells, including various neurons and glial cells, as well as extracellular supportive matrices [12]. Therefore, in the event of a stroke where many neuronal tissues are damaged, a sudden release of CNS and/or vascular biomarkers into the peripheral blood would be expected. If such biomarkers are reliably measured in the peripheral blood specimens, then they could be used for screening or triaging purposes.

In this review, we have mentioned many currently available stroke biomarkers but have explained seven single blood biomarkers and five biomarker panels in more detail because of their potential usefulness for the detection or prediction of IS in suspected patients.

## 2. Materials and Methods

We performed a narrative review of the literature published in the English language from 2001 to 2021 using two online databases, PubMed and Web of Science. We used the search terms “stroke”, “diagnosis”, “biomarker”, “humans”, “sensitivity”, and “specificity”. We also screened the reference lists of the extracted articles to identify articles not computed from the original search.

## 3. Results and Discussion

The initial database search generated 170 results. Three articles were excluded as duplicates, and after screening the titles and abstracts of the remaining, we included 23 articles for this review (Figure 1 and Table 1).

### 3.1. Individual Biomarkers

Over the last 20 years, many biomarkers have been studied for stroke diagnosis; however, we still do not have a reliable biomarker that can detect stroke with a high accuracy compared to troponin in the diagnosis of ischaemic heart disease. Nonetheless, so far, many biomarkers have been identified whose blood levels increase following a stroke event, especially an acute IS. In general, those biomarkers can be divided into a few categories based on their origins, namely: (1) the neuronal injury markers (e.g., heart-type fatty acid binding protein (H-FABP), NR2 peptide (a degradation product of N-methyl-d-aspartate receptors found in plasma), Parkinson disease protein 7 (PARK7), nucleoside diphosphate kinase A (NDKA), apolipoprotein A1 unique peptide (APOA1-UP), matrix metalloproteinase-9 (MMP-9), glycogen phosphorylase isoenzyme BB (GPBB), and B-type neurotrophic growth factor (BNGF)) [13,15,16,17]; (2) the neuronal cell activation indicators (e.g., S100 calcium-binding protein B (S100B) and monocyte chemoattractant protein-1 (MCP-1)) [18,19]; (3) the neuroinflammation indicators (e.g., eotaxin and vascular cell adhesion molecule (VCAM)) [20,21]; (4) the endothelial dysfunction markers (e.g., D-dimer, von Willebrand factor (vWF); and (5) the neuro-endocrine markers such as B-type natriuretic peptide (BNP), and cortisol [22].

Despite the abundance of available biomarkers, only a few of them have demonstrated a sensitivity above 50% for stroke in clinical trials, which largely limits their clinical applicability [23]. In the process of this literature review, we focused on biomarkers that have undergone preliminary clinical evaluations. We selected seven individual biomarkers that have both sensitivity and specificity of more than 50% (Figure 2).

S100B is a member of the S100 protein superfamily. It is an intracellular protein found in glial cells and Schwann cells and is released into the blood circulation following cellular activation caused by tissue damage [24,25]. Zhou et al., (2016) reported a sensitivity of 95.7% and specificity of 70.4% for stroke for S100B, as well as an area under the curve (AUC) of 0.903 in differentiating between IS and intracranial haemorrhage (ICH) [18]. In another study, this biomarker was found useful in predicting the patient’s short-term functional outcome after a stroke event [24]. However, the elevations of the plasma levels of this biomarker in other neurological and neuropsychological disorders such as Alzheimer’s disease and schizophrenia means it would be of a reduced value in triaging the suspected patients for stroke [25].

GPBB is a glycogen phosphorylase isoenzyme found in the brain and heart tissues whose function is to make glucose-1-phosphate by breaking down glycogen, which helps restore the energy stores, which are depleted during a cerebral ischaemic event [17]. According to Park et al., (2018), increased plasma levels of GPBB have a sensitivity and a specificity of 93% for detecting stroke within 12 h from the onset of the symptoms [17]. However, this study did not find any correlation between GPBB levels and the severity of the stroke, infarct volume, or the clinical outcome, which suggests a less suitable position for this biomarker to be used for predicting the disease prognosis in patients with IS.

NR2 peptide is an N-terminal fragment of N-methyl-D-aspartate (NMDA) receptors, which can be measured in the plasma sample. Following cerebral ischaemia, NMDA receptors are released from endothelial cells of the brain’s microvessels which are then cleaved by serine proteinases and are released into the blood stream as NR2 peptides [14]. In a study undertaken by Dambinova et al., in 2012, it was reported that NR2 peptide has a sensitivity of 92% and specificity of 96% for ischaemic stroke when measurable at 3 h post-stroke [14]. Also, it has been found that the plasma levels of anti-NMDA antibodies are predictive of stroke and a brain lesion size in high-risk patients [26].

MMP-9 is a Zn^2+^-dependent proteolytic enzyme that is released from different cells such as neutrophils and has roles in the degradation of the extracellular matrix following IS and ICH [36]. Experimental studies have shown that systemic inflammation during stroke causes a neutrophil infiltration of the ischaemic area of the brain which eventually leads to increased plasma MMP-9 activity in patients with stroke [27]. The studies by Castellanos et al., (2007) and Kelly et al., (2008) showed that high levels of MMP-9 are predictive of blood–brain barrier disruption and haemorrhagic transformation after an IS [28,29]. Another benefit of measuring plasma MMP-9 has been reported to be its predictive value in detecting brain tissue haematoma following tissue plasminogen activator treatment in patients with IS (sensitivity of 92% and specificity of 74%) [29].

Apolipoprotein A1 (APOA1) is a major protein component of the high-density lipoprotein (HDL) and exhibits anti-inflammatory and antioxidant effects, hence playing an important role in the protection of the vascular system against oxidative stress. Studies have shown that the levels of APOA1 decrease in patients with stroke and/or infection [30]. Similarly, decreased levels of APOA1-UP, as a novel biomarker have been reported to have a high sensitivity (91%) and specificity (97%) for the prediction of IS, nominating it as a promising independent predictor of IS [16].

PARK7 and NDKA are released from the cerebrospinal fluid into the plasma after significant brain injury [15]. The study done by Allard et al., (2005) reported a sensitivity of 54% and a specificity of 90% for PARK7 at a cut-off level of 14.1 μg/L, when samples were taken at 3 h after the onset of the acute stroke. Accordingly, the reported sensitivity and specificity for NDKA, at a cut-off value of 22 μg/L were 70% and 90%, respectively [15].

H-FABP is a fatty acid binding protein that is released from CNS tissues after an ischaemic event into the blood. A study by Park et al., found that this protein had a sensitivity of 59.5%, specificity of 79.5%, and an AUC = 0.71 (*p* < 0.001) for identifying IS if the blood samples were collected after 24 h of the stroke onset [13]. Given the long timeframe and its low sensitivity, this protein might not be a good biomarker for stroke detection.

Although many of these biomarkers seem promising in the early screening of stroke, most of the findings are hardly generalisable to larger populations due to the small sample sizes of the original studies. In addition, because medical interventions need to be performed within a short timeframe to salvage the vulnerable neuronal tissues and minimise mortality or functional deficits, many of the suggested biomarkers do not seem to be very useful because of the relatively long time needed for the symptoms’ onset until a reliably measurable change in the biomarkers’ levels can be detected. Some biomarkers, such as PARK7, NDKA, and NR2 peptide, are released into the plasma and are detectable within the first three hours after the stroke onset, which makes them potentially promising biomarkers to be used in future studies in acute settings [15]. Unfortunately, many other biomarkers identified in this review have not yet been evaluated for their diagnostic reliability at the early stages of stroke. Table 2 summarises some of the key aspects of the clinical studies related to the biomarkers and biomarker panels reviewed in this article.

As it can be seen from Figure 2, some of these biomarkers (e.g., S100B) have better sensitivity than others, but are less specific for stroke [37]. In addition, some comorbidities and other factors are also found to interfere with the accuracy of these biomarkers. However, when the biomarkers are combined in a panel, they may offer greater sensitivity and specificity values compared to individual biomarkers [37].

### 3.2. Biomarker Panels

Unavailability of single biomarkers with both high sensitivity and specificity has been a limiting factor in adopting blood biomarkers as stand-alone diagnostic tools in clinical situations such as stroke. To add to the complexity, patterns of biomarkers changes may differ depending on the type of stroke (e.g., IS versus ICH) or depending on the affected brain areas [31]. It has been suggested that by combining several biomarkers into a biomarker panel more useful information can be obtained particularly by including biomarkers specific to different areas of the brain [14,16,17,29]. In this review, we have identified five biomarker panels that have shown both a sensitivity and specificity above 50% (Figure 3). We have named these five biomarker panels as panels A through E in this review due to the lack of specific trade names for them in the original articles (Table 2 and Table 3).

Most of these panels are composed of brain-specific biomarkers (neuronal cell activation and neuro-endocrine markers) and non-specific biomarkers (MMP-9, C-reactive protein (CRP), VCAM, vWF, and D-dimer), to represent different parts of the ischaemic cascade and provide complementary information for the diagnosis of stroke. Although the findings from those studies are not conclusive, the use of biomarker panels may have opened a new frontier in the development of highly sensitive and specific biomarkers. Therefore, the concept of diagnostic biomarker panels is a promising topic for future research.

Panel A is composed of five protein biomarkers that were studied by Reynolds et al., (Table 2) [32]. This panel has shown a sensitivity of approximately 98% and specificity of about 93% for prediction of IS for samples collected within 6 h from the appearance of symptoms. This is a significant improvement compared to many individual markers in previous studies [32]. Panel B includes three protein biomarkers based on a study by Lynch et al., in 2004. This panel had both 90% sensitivity and specificity where the samples were obtained within 6 h of the stroke onset [33]. Panel C comprises five protein biomarkers based on a study by Sharma et al., in which they reported a sensitivity of 90%, specificity of 84%, a positive predictive value (PPV) of 78%, and a negative predictive value (NPV) of 93% for stroke detection within 24 h of symptoms’ onset [20]. Panel D, which was developed in a cohort of 130 patients with acute neurological symptoms, consists of five protein biomarkers. This panel showed a sensitivity of 81% and a specificity of 70% for the prediction of IS when the blood specimens were collected within 6 h of the stroke onset [35]. Given the above information, panels A, B, and D may be clinically useful for triaging purposes [32,33,35].

Panel E, made by Moore et al., in 2005 was created following a comparative study of gene expression profiles in confirmed stroke cases (IS; n = 20) versus matched healthy controls (n = 20) using microarray technology. Accordingly, and after the initial study of exhaustive gene expression patterns using the RNA samples extracted from peripheral blood mononuclear cells, they observed a significant change (mainly up-regulations) in the expression of 190 genes in patients with IS. Next, a panel of 22 genes was chosen for the derivation of a predictive model for the prediction of stroke using hierarchical cluster analysis (Table 3). The model was then prospectively validated in another cohort consisting of 9 stroke patients and 10 healthy individuals. This model showed a sensitivity of 78% and a specificity of 80% in the validation cohort [35]. These results are promising; however, the authors were unable to rule out the effects of non-stroke causes in the up-regulation of the genes. In addition, because of the small sample sizes both for the derivation and the validation studies, the results need to be validated in larger studies.

The above-mentioned biomarkers have not been approved for clinical diagnostic use yet due to several reasons, including the lack of large prospective trials, lack of the standards for measurement, unknown interference in certain population groups, or uncertainties in the time-concentration relationships. We believe that the available data are still limited, and explorative investigations such as in vitro studies on stroke biomarkers are still insufficient. We suggest that before starting large-scale clinical studies, we need to have a better understanding of the window periods of individual biomarkers for stroke detection (the time from symptoms’ onset to a detectable change in the blood levels of a biomarker). As we know, the efficacy for current interventions for acute stroke is time-dependent, and most of the current guidelines recommend <4.5 h as a key target between the onset of symptoms and treatment intervention with fibrinolytic, and up to 6 h for mechanical thrombectomy [38,39]. Therefore, by taking into consideration the time required for radio-imaging to confirm the diagnosis prior to treatment (which is around 45 min in optimal settings and up to 1.5 h in average settings), any biomarker that can be reliably detected within 3 h of the onset of stroke could be a highly valuable diagnostic tool.

In this literature review, we did not allocate a great level of priority for exploring potential associations between blood levels of biomarkers and the size of brain lesions (extents of infarcts or bleedings) as our review was more of a diagnostic nature rather than prognostic. However, we found some limited, yet promising, evidence, which could be used in future research in those areas. For example, in terms of the IS, Kalev-Zylinska et. al. (2013) found that plasma levels of anti-NMDA antibodies were predictive of brain lesion’s size in the patients and a high National Institutes of Health Stroke Scale score. However, they found this association only in 21 of the 48 patients (44%) studied. Accordingly, they identified this small sample size and the explanatory nature of their research as limiting factors for the applicability of their findings [26]. On the other hand, in a study by Park et al., (2018), the researchers did not find a correlation between GPBB levels and the clinical outcome or volume of the infarct in patients with IS [17]. In addition, in terms of the intracerebral haemorrhage, there is some promising evidence. For example, in 2007 Castellanos et al., in a multicentre prospective study reported high sensitivities and negative predictive values for the levels of serum cellular fibronectin and MMP-9 for the prediction of haemorrhagic transformation and parenchymal haematoma following thrombolytic therapy in patients with acute IS. However, they admitted that their small sample size (n = 134) and the smaller number of patients who developed parenchymal haemorrhage during the follow-up period were among the limiting factors for their findings. Having mentioned all the above, and despite the uncertainties in the usefulness of biomarkers in predicting the lesion size and clinical outcomes, this is a relatively less explored area of knowledge, and we believe that it is worth researching further.

## 4. Our Study Limitations

There are a few limitations to this paper. Firstly, we performed a narrative literature search using studies involving human only and excluded animal studies, which may have caused us to miss some of the current literature. Secondly, this was not a systematic review; therefore, we are not sure if we have identified and reported all the appropriate stroke biomarkers (hence we suggest a systematic review for this purpose). Thirdly, we only searched for articles published in English; as a result, we may have missed some important studies published in non-English languages. Lastly, because most of the patients in the included studies were middle-aged or older adults, some of the conclusions presented here might not be applicable to children and young adults because of age-related differences in the pathophysiology of stroke. Accordingly, we suggest that there is an urgent need for research into the role of blood biomarkers in the detection of stroke in different age groups, particularly children and young adults.

## 5. Conclusions

The results of this literature review indicate that there are potential biomarkers (both as individual biomarkers and as panels) with high-enough sensitivity and specificity that may serve as early detection tools for stroke diagnosis. However, most of the published studies had small sample sizes, which makes the clinical applicability of their findings challenging. Therefore, further research needs to be done in larger cohorts to confirm the clinical usefulness of the available data. In addition, most of the proposed biomarkers have not been examined in very acute patients within the first 3–5 h post-stroke, hence there is a need for further research in this area.

## Figures and Tables

**Figure 1 jcm-11-04243-f001:**
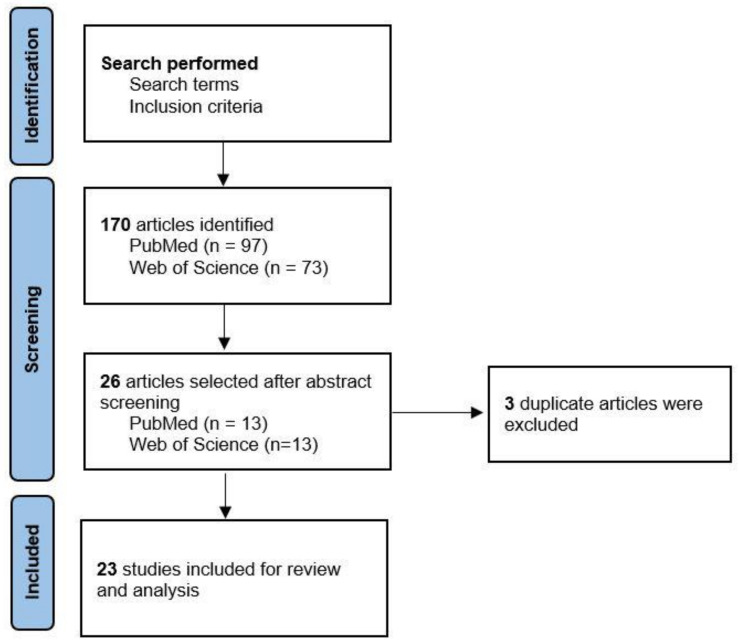
The literature search process.

**Figure 2 jcm-11-04243-f002:**
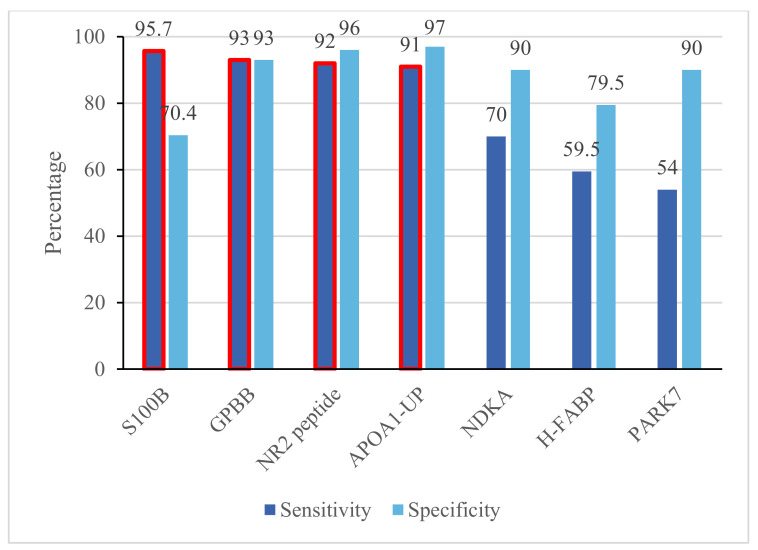
Single blood biomarkers for stroke. S100B, GPBB, NR2 peptide, and APOA1_UP all have sensitivities of >90% (highlighted in red). S100B, S100 calcium-binding protein B; GPBB, glycogen phosphorylase BB; NR2 peptide, a degradation product of N-methyl-d-aspartate receptors; APOA1-UP, apolipoprotein A1 unique peptide; NDKA, nucleoside diphosphate kinase A; H-FABP, heart-type fatty acid binding protein; PARK7, Parkinson disease protein 7.

**Figure 3 jcm-11-04243-f003:**
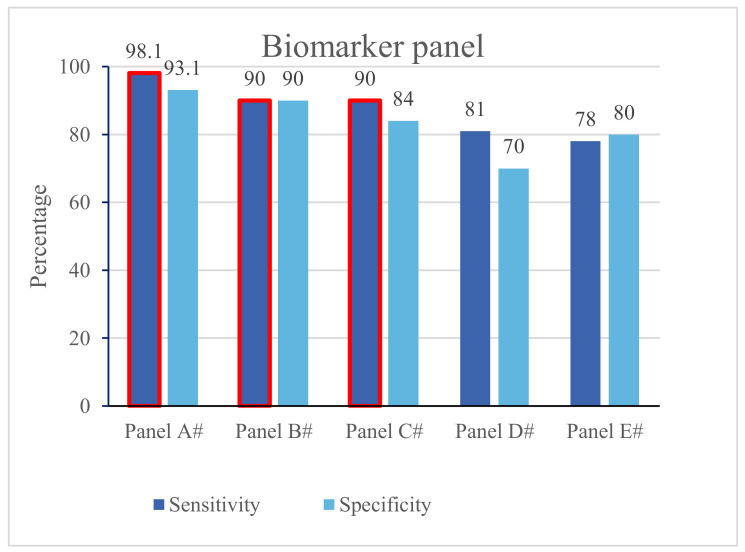
Panel biomarkers for stroke. Panels A, B and C have sensitivity of 90% or higher, which are highlighted in red. NB: The reported sensitivity and specificity for panels A, B, D, and E are related to ischaemic stroke only.

**Table 1 jcm-11-04243-t001:** Included studies in this literature review.

Author/s (Year), Reference Number	Type of Study, Country	Numbers of Participants and Controls, Mean or Median Age (Age Range When Available)	Type of Stroke	Biomarkers/Biomarker Panels Studied
Park S. Y., et al., (2013) [13]	Cohort study, Korea	Patients: n = 111, mean age 67; controls: n = 127, mean age 63	IS	H-FABP and S100B
Dambinova S. A., et al., (2012) [14]	Cohort study, USA	Patients: n = 101, median age 62 (26–95); non-stroke patients (stroke mimics): n = 91, median age 61 (24–95); healthy controls: n = 52, median age 59 (29–92)	IS or TIA	NR2 peptide
Allard L., et al., (2005) [15]	Cohort studies; one European (Switzerland) and two American (USA) cohorts.	European study: patients: n = 36, mean age 71.3 (25–92); controls: n = 35 mean age 71.1 (28–91); American study 1: patients: n = 53, controls: n = 30 (non-age or sex-matched); American study 2: patients: n = 533, controls: 100 (age matched with patients).	IS (most of the patients), TIA, HS	PARK7 and NDKA
Zhao X., et al., (2016) [16]	Cohort study, China	Patients: n = 94, mean age 61.8; controls: n = 37, mean age 47.1	IS	APOA1-UP
Park K. Y., et al., (2018) [17]	Cohort study, USA	Patients: n = 172, mean age 68.8; controls: n = 133, mean age 71.0	IS	GPBB
Zhou, S. et al., (2016) [18]	Single-centre pilot study, China	Patients: HS: n = 46, mean age 68.1; IS: n = 71, mean age 69.3; no control group	HS and IS	S100B
Losy, J. and Zaremba, J. (2001) [19]	Cohort study, Poland	Patients: n = 23, mean age 72.2; controls: n = 15 (age and sex-matched)	IS	MCP-1
Sharma, R., et al., (2014) [20]	Cohort study, USA	Patients: IS: n = 56, mean age 66.9; HS: n = 32, mean age 64.7; TIA: n = 41, mean age 63.1; mimic: n = 37, mean age 61.8	Mixed patient group (IS, HS, TIA, and mimics)	A 5-biomarker model was developed consisting of eotaxin, EGFR, metalloproteinase inhibitor-4, prolactin, and S100A12
Supanc, V., et al., (2011) [21]	Cohort study, Croatia	Patients: n = 110; mean age 70.2 (36–86); controls: n = 93, median age 70 (47–86)	IS	ICAM-1 and VCAM-1
Katan, M., Elkind, M. S. V. (2011) [22]	Review article, USA	N/A	IS	IL-1, IL-6, MM-9, TNF-alpha, TNF-a receptor, ICAM-1, VCAM, Lipoprotein-associated phospholipase A2, vWF; Fibrinogen; D-dimer, BNP, NT-proBNP, cortisol; PAI-1, and others
Kamtchum-Tatuene, J. and Jickling, G. C. (2019) [23]	Review article, Canada	N/A	IS and HS	S100B, GFAP, MBP, NSE, H-FABP, anti-NMDA receptors antibodies, vWF, D-dimer, fibrinogen, PAI, Fibronectin, MMP-9, caspase-3, thrombomodulin, and others
Abdel-Ghaffar, W. E. et al., (2019) [24]	Cohort study, Egypt	Patients: n = 40, age above 65 years old; no control group	IS and HS	S100B
Sen, J. and Belli, A. (2007) [25]	Review article, UK	N/A	N/A	S100B
Kalev-Zylinska, M. L. et al., (2013) [26]	Cohort study, New Zealand	Patients: n = 48, Mean age 70; control group 1: health laboratory workers: n = 46, age range 30 years of age or younger; control group 2: healthy blood donors: n = 50, age range 50 years of age or older	IS	Anti-NMDAR antibodies
Lakhan S. E. et al., (2013) [27]	Review article, USA	N/A	IS	MMP-9
Kelly, P. J. et al., (2008) [28]	Case–control study, Ireland	Patients: n = 52; mean age 70.1; controls: n = 27, mean age 68.2	IS	MMP-9 and F2Ips
Castellanos, et al., (2007) [29]	Cohort study, Spain	Patients: n = 134, mean age 62; no control group	IS	MMP-9
Eldeeb, M. A. et al., (2020) [30]	Case–control, Egypt	Patients: n = 60, mean age 60, age range 28–88; healthy controls: n = 30 (age and sex-matched)	IS	Apo-A1
Kawata, K. et al., (2016) [31]	Review article, USA	N/A	IS and HS	S100B, NSE, MMP-9, sCD40L, TIMP-1, MDA, and others
Reynolds, M. A. (2003) [32]	Cohort study, USA	Patients: n = 223 (including 82 patients with IS), age not available; controls (healthy donors): n = 214, age not available	IS and HS (a mixed patient group)	A 5-biomarker panel was developed consisting of S100B, BNGF, vWF, MMP-9, MCP-1,
Lynch, J. R. et al., (2004) [33]	Cohort study, USA	Patients: n = 65, mean age 62; controls (non-stroke): n = 157, mean age 63.3	IS	A 3-biomarker panel was developed consisting of vWF, MMP-9, and VCAM
Laskowitz, D. T. et al., (2005) [34]	Cohort study, USA	Patients: n = 130, age not available; controls: n = 10, age not available	IS	A 5-biomarker panel was developed using BNP, CRP, D-dimer, MMP-9, and S100B.
Moore, D. F. et al., (2005) [35]	Cohort study, Canada	Patients (IS): n= 20, mean age 75.5; controls (healthy): n = 20, mean age 66.0	IS	A 22-gene expression panel was developed using peripheral blood mononuclear cells.

H-FABP, heart-type fatty acid binding protein; S100B, S100 calcium-binding protein B; TIA, transient ischaemic attack; NR2 peptide, NMDA (N-methyl-d-aspartate) receptor 2 peptide; PARK7, Parkinson disease protein 7; NDKA, nucleoside diphosphate kinase A; APOA1-UP, apolipoprotein A1 unique peptide: GPBB, glycogen phosphorylase BB; IS, ischaemic stroke; HS, haemorrhagic stroke; MCP-1, monocyte chemoattractant protein-1; EGFR, epidermal growth factor receptor; S100A12, S100 calcium-binding protein A12; ICAM-1, inter-cellular adhesion molecule 1; VCAM-1, vascular cell adhesion molecule 1; IL-1, interleukin 1, IL-6, interleukin 6; MMP-9, matrix metalloproteinase-9; TNF: tumour necrosis factor; vWF, von Willebrand factor; BNP, brain natriuretic peptide; NT-pro BNP, N-terminal pro-brain natriuretic peptide; PAI-1, plasminogen activator inhibitor-1; GFAP, glial fibrillary acid protein; MBP, myelin basic protein; NSE, neuron-specific enolase; Anti-NMDAR, antibody against N-methyl-d-aspartate receptor; F2Ips, F2-isoprostanes; Apo-A1, apolipoprotein A1; NSE, neuron-specific enolase; sCD40L, soluble CD40 ligand; TIMP-1, tissue inhibitors of metalloproteinases-1; MDA, malondialdehyde; BNGF, B-type neurotrophic growth factor; MCP-1, monocyte chemotactic protein-1.

**Table 2 jcm-11-04243-t002:** Blood biomarkers for stroke diagnosis.

Biomarker	Reference	Sample Size (n)	Cut-Off	Time from Symptoms Onset to Sample Collection (up to)
S100B	Zhou et al., (2016) [18]	46 (ICH) 71 (IS)	67 pg/mL	6 h
GPBB	Park et al., (2018) [17]	172 (IS) 133 (C)	7.0 ng/mL	4.5 h
NR2 peptide	Dambinova et al., (2012) [14]	101 (IS) 91 (C)	1.0 μg/L	3 h
APOA1-UP	Zhao et al., (2016) [16]	94 (IS) 37 (C)	APOA1-UP/LRP ratio 1.80	72 h
PARK-7	Allard et al., (2005) [15]	622 (S) 165 (C)	9.33 μg/L	3 h
NDKA	Allard et al., (2005) [15]	622 (S) 165 (C)	2 μg/L	3 h
H-FABP	Park et al., (2013) [13]	111 (IS) 127 (C)	9.70 ng/ml	24 h
Panel A	Reynolds et al., (2003) [32]	223 (S) 214 (C)	-	6 h
Panel B	Lynch et al., (2004) [33]	65 (IS) 157 (C)	-	6 h
Panel C	Sharma et al., (2014) [20]	167 (S)	-	24 h
Panel D	Laskowitz et al., (2005) [34]	130 (IS) 10 (C)	-	6 h
Panel E	Moore et al., 2005 [35]	20 (IS) 20 (C)	-	<24 h (n = 7), 24–48 h (n = 10), >48 h (n = 3)

Ischaemic stroke (IS), control (C), intracerebral haemorrhage (ICH), stroke (not specified or a mixed population) (S), labelled reference peptide (LRP).

**Table 3 jcm-11-04243-t003:** Panel biomarker composition.

Biomarker Panel	Composition of Biomarkers
Panel A (5 proteins)	BNGF, MCP-1, MMP-9, S100B, vWF
Panel B (3 proteins)	vWF, MMP-9, VCAM
Panel C (5 proteins)	Eotaxin, EGFR, S100A12, Metalloproteinase inhibitor-4, Prolactin
Panel D (5 proteins)	S100B, MMP-9, D-dimer, BNP, CRP
Panel E (22 genes)	CD163; Hypothetical protein FLJ22662 Laminin A motif; Amyloid β(A4) precursor-like protein 2; *N*-acetylneuraminate pyruvate lysase; *v-fos* FBJ murine osteosarcoma viral oncogene homolog; Toll-like receptor 2; Ectonucleoside triphosphate diphosphohydrolase 1; Chondroitin sulfate proteoglycan 2 (versican); Interleukin 13 receptor, α1; CD14 antigen; Bone marrow stromal cell antigen 1/CD157; Complement component 1, q subcomponent, receptor 1; Paired immunoglobulin-like type 2 receptor α; Fc fragment of IgG, high-affinity Ia, receptor for (CD64); Adrenomedullin; Dual-specificity phosphatase 1; Cytochrome b-245, β polypeptide (chronic granulomatous disease); Leukotriene A4 hydrolase; *v-ets* Erythroblastosis virus E26 oncogene homolog 2 (avian); CD36 antigen (thrombospondin receptor); Baculoviral IAP repeat-containing protein 1 (Neuronal apoptosis inhibitory protein); and KIAA0146 protein

BNGF, B-type neurotrophic growth factor; MCP-1, monocyte chemoattractant protein-1; MMP-9, matrix metalloproteinase-9; S100B, S100 calcium-binding protein B; vWF, von Willebrand factor; VCAM, vascular cell adhesion molecule; EGFR, epidermal growth factor receptor; S100A12, S100 calcium-binding protein A12; BNP, B-type natriuretic peptide; CRP, C-reactive protein.

## Data Availability

No data were generated in this study.

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
