# Peer review of "Blood Biomarkers for Triaging Patients for Suspected Stroke: Every Minute Counts"

_jcm, 2022, doi:10.3390/jcm11144243_

Round 1

Reviewer 1 Report

This review article provided comparative information on the availability, clinical usefulness, and time-window periods of eight single blood biomarkers and six biomarker panels that have been used for predicting stroke in emergency situations. This study may be used in future research for developing effective stroke biomarkers. The authors summarized several articles well and then described them in a concise manner. Here are my suggestions:

  1. The inclusion/exclusion criteria are ambiguous. What is the PICO Criteria for this study?
  2. Which of the authors did the article review/selection? How did you resolve disagreements that occurred during the selection process?
  3. Please provide a table of information on the 23 papers included in the final study.
  4. The authors stated that the specificity of S100B for distinguishing between ischemic and hemorrhagic stroke was over 90%. That is, S100B is presumed to be a diagnostic marker for ischemic stroke and not hemorrhagic. The reason I mentioned is that blood markers can vary widely depending on the type and subtype of stroke. In this regard, authors should present the final stroke type of subjects included in the references.
  5. Were there any studies that reported the efficacy of blood markers according to stroke subtype?
  6. Were there any blood markers that showed dose effects with infarct size or cerebral hemorrhage? Please include additional information about them in the discussion.
  7. The title suggests stroke blood markers for young adults and children. However, age-related associations and validity for these markers are lacking. Therefore, the title should be changed or the discussion about young strokes should be augmented.

Author Response

Response to Reviewer 1 Comments

Point 1: The inclusion/exclusion criteria are ambiguous. What is the PICO Criteria for this study?

Response 1:

Thank you for your comment. We understand that PICO is a highly valuable criteria in identifying a narrow and specific research question for doing a systematic or scoping review. However, our research was a narrative literature review not a systematic review; we therefore clarified this matter on page 2, line 74 and page 13, line 319. We undertook our literature review to provide a general overview of the literature about the availability and usefulness of blood biomarkers in traiging suspected patients for stroke. Accordingly, we intended to keep the scope of our study  broad, so we believe we did not need to define a narrow research question using PICO.

Point 2: Which of the authors did the article review/selection? How did you resolve disagreements that occurred during the selection process?

Response 2:

Thanks for the comment. In general, all authors were involved in all aspects of the manuscript development, although to different extents. Considering that this work is a narrative literature review (and not a systematic review), we believe that some of the requirements for systematic reviews, such as an exact description of conflict resolutions, do not apply here. However, we have addressed this question in the “Author Contributions” section in page 13, lines 341 – 345.

Point 3: Please provide a table of information on the 23 papers included in the final study.

Response 3:

Thank you for highlighting this important issue. Actually, we believe this comment helped us improve the quality of our paper to a great extent (so, we really thank you for that). We have addressed this comment in the manuscript by adding a new table to it (Table 1, pages 3 to 7).  

Point 4: The authors stated that the specificity of S100B for distinguishing between ischemic and hemorrhagic stroke was over 90%. That is, S100B is presumed to be a diagnostic marker for ischemic stroke and not hemorrhagic. The reason I mentioned is that blood markers can vary widely depending on the type and subtype of stroke. In this regard, authors should present the final stroke type of subjects included in the references.

Response 4:

We thank the reviewer for this comment. We carefully revised our manuscript and provided more details about the types of stroke when it comes to diagnostic sensitivity and specificity of S100B and other biomarkers (where the data were available from the original manuscripts). We have applied this pattern throughout our manuscript now. Based on the available data, no single blood biomarker is accurate enough to be used for the differentiation of stroke types or subtypes. The primary methods to confirm the type, subtype, location, and the severity of stroke still depend on the radio-imaging technology. However, biomarkers may be a useful tool in triaging patients for rapid radio-imaging tests.

Point 5: Were there any studies that reported the efficacy of blood markers according to stroke subtype?

Response 5:

No, the greatest level of details in the included articles was to refer to the types of stroke (ischemic stroke vs haemorrhagic stroke) not the subtypes of stroke. We, therefore, followed the patterns found in the included studies.

Point 6: Were there any blood markers that showed dose effects with infarct size or cerebral hemorrhage? Please include additional information about them in the discussion.

Response 6:

Thank you for the comment. We added a whole new paragraph (pages 12 and 13, lines 294 – 316) to address the comment.

Point 7: The title suggests stroke blood markers for young adults and children. However, age-related associations and validity for these markers are lacking. Therefore, the title should be changed or the discussion about young strokes should be augmented.

Response 7:

Thanks for the comment. We agree with the reviewer that we have not discussed the effectiveness of using the blood biomarkers in children and young adults to a great extent (probably due to the scarcity of published resources). We therefore changed the manuscript’s title to cater for the whole patient population for the stroke (page 1, lines 2 and 3). Having said that, we still believe that a need for clinically useful biomarkers for stroke is even more important in children and young adults because of the relative abundance of stroke mimics among those populations. As a result, we kept in place the descriptions regarding the importance of such triaging tools among children and young adults.

Reviewer 2 Report

Thank you for possibility to read and review your manuscript. The topic you have chosen is particularly important and relevant for the differential diagnosis and emergency treatment of acute stroke. The manuscript is well written and easy to read. I have very few comments and suggestions for improving the manuscript.

1. Page 7, line 214: "specifically" probably should be changed to "specificity".

2. Page 8, lines 248-250: " As we know, the efficacy for current interventions for acute stroke is time-248 dependent, and most of the current guidelines recommend <6.5 hours as key target between the onset of symptoms and treatment intervention.". The majority of guidelines still recommend 4,5 h therapeutic window for intravenous thrombolysis in AIS.I tend to think that 6.5 hours is too long from the first symptoms of a stroke to the start of stroke treatment. In order for stroke biomarkers to be of real clinical significance, one should focus on a shorter time interval.

3. The importance of searching for blood biomarkers or panels of biomarkers for acute stroke is not limited to triaging young adults and children. This is important when a stroke is suspected at any age. Therefore, is it necessary to emphasize the young stroke contingent in the manuscript title? However, this is the prerogative of the authors.

Thank you.

Author Response

Responses to Reviewer 2 Comments

Point 1: Page 7, line 214: "specifically" probably should be changed to "specificity".

Response 1:

We thank you for highlighting this issue. You are correct. We changed “specifically” to “specificity”. It is now in page 12, line 251.

Point 2: Page 8, lines 248-250: " As we know, the efficacy for current interventions for acute stroke is time-248 dependent, and most of the current guidelines recommend <6.5 hours as key target between the onset of symptoms and treatment intervention.". The majority of guidelines still recommend 4,5 h therapeutic window for intravenous thrombolysis in AIS.I tend to think that 6.5 hours is too long from the first symptoms of a stroke to the start of stroke treatment. In order for stroke biomarkers to be of real clinical significance, one should focus on a shorter time interval.

Response 2:

Thank you for this important comment. We agree to that, and we made the required changes as follows:

  • Changed 6.5 hours to 4.5 hours as the preferred optimal time for thrombolysis), and specify 6 hours for mechanical thrombectomy as per 2019 AHA/ASA Guideline (page 12, line 287.
  • We also added a new reference to address this matter (page 12, line 289). We changed our suggested timeframe for the usefulness of ideal blood biomarkers from 5 hours to 3 hours (from the start of the symptoms). The change can be seen on page 12, line 292.

Point 3: The importance of searching for blood biomarkers or panels of biomarkers for acute stroke is not limited to triaging young adults and children. This is important when a stroke is suspected at any age. Therefore, is it necessary to emphasize the young stroke contingent in the manuscript title? However, this is the prerogative of the authors.

Response 3:

We appreciate your comment and agree to it. Accordingly, we deleted the phrase “young adults and children” from the title of our manuscript so that it can reflect the usefulness of the availability of biomarkers to all age groups. However, due to the existence of more diagnostic challenges in children and young adults, we maintained our emphasis in the text on the importance of the availability of such biomarkers for children and young adults with suspected stroke.

Reviewer 3 Report

In the submitted manuscript entitled "Blood biomarkers for triaging young adults and children for a 2 suspected stroke: every minute counts. Review" by the group of Faculty of Health, University of Canberra, Canberra 2617, ACT, Australia, the authors A made a review on PubMed and Web of Science for 20 journal articles published in English during the period 2001 to 2021 which contained information 21 regarding biomarkers of stroke. They provided comparative information on the 22 availability, clinical usefulness, and time-window periods of eight single blood biomarkers and six 23 biomarker panels that have been used for predicting stroke in emergency situations. Although there are obvious limitations in their manuscript due to the fact that the search included mainly adult and aged populations, and this article deals with the disease in children and adolescents, and also that there is indeed "much room" to accommodate major prospective multicenter initiatives focusing in the particular setting of their outcomes, the reviewer thinks that the authors should be congratulated with a nicely written manuscript in a somehow original subject.

Major comments: None.   

Good luck.

Author Response

We would like to thank you for the review of our manuscript and your encouraging words. We appreciate that our work, as a narrative literature review, has its limitations, but as you have highlighted, we have been able to gather some good evidence about the clinical usefulness of single or panel biomarkers in triaging patients for stroke (especially ischaemic stroke), and have also identified some areas for future research. We thank you again for reviewing our manuscript.

Round 2

Reviewer 1 Report

Thank you.